Accepted at the ICLR 2024 Workshop on AI4Differential Equations In Science

# OPTIMAL EXPERIMENTAL DESIGN FOR BAYESIAN IN­VERSE PROBLEMS USING ENERGY-BASED COUPLINGS

**Paula Cordero Encinar, Tobias Schröder & Andrew B. Duncan**
Department of Mathematics
Imperial College London
London , UK
{paula.cordero-encinar22,t.schroeder21,a.duncan}@imperial.ac.uk

## ABSTRACT

Bayesian Experimental Design (BED) is a robust model-based framework for op­timising experiments but faces significant computational barriers, especially in the setting of inverse problems for partial differential equations (PDEs). In this pa­per, we propose a novel approach, modelling the joint posterior distribution with an energy-based model, trained on simulation data. Unlike existing simulation-based inference approaches, we leverage implicit neural representations to learn a functional representation of parameters and data. This is used as a resolution-independent plug-and-play surrogate for the posterior, which can be conditioned over any set of design-points, permitting an efficient approach to BED.

## 1 INTRODUCTION

Mathematical models based on partial differential equations (PDEs) play a crucial role in quan­titative analysis of complex systems arising in science, engineering and socio-economics (Burger et al., 2014). These often involve large numbers of parameters which must be calibrated. In many settings, the unknown parameters are functions, e.g. spatially-varying coefficient fields. Identify­ing parameters based on measurement data involves solving an *inverse problem*: Given a possibly stochastic forward operator $\mathcal{G} : \mathcal{A} \to \mathcal{U}$ between a parameter space $\mathcal{A}$ and a solution space $\mathcal{U}$, and empirical observations of the solution $\mathbf{y} \in \mathbb{R}^n$ at locations $\underline{\mathbf{x}} = (\mathbf{x}_1, \ldots, \mathbf{x}_n), \mathbf{x}_i \in \Omega \subset \mathbb{R}^d$, we aim to infer the parameter $a \in \mathcal{A}$ such that $\mathbf{y} = \mathcal{G}(a)(\underline{\mathbf{x}}) + \eta$, where $\eta$ denotes mean-zero obser­vational noise. The Bayesian approach to inverse problems (Stuart, 2010) is a systematic method for uncertainty quantification of parametric estimates, where $a$ and $\mathbf{y}$ are viewed as coupled random variables. Equipping $a \in \mathcal{A}$ with a prior distribution, Bayes' rule yields a posterior distribution for $a$ given $\mathbf{y}$.

A significant challenge is determining which measurement positions $\underline{\mathbf{x}}$ yield the most information about an unknown parameter. This question can be tackled using Bayesian Experimental Design (BED) (Chaloner & Verdinelli, 1995), a versatile framework that guides the process of data col­lection. Typical approaches to BED involve a nested Monte Carlo approach, with Markov Chain Monte Carlo (MCMC) methods used to estimate a utility for any given candidate design, which is subsequently optimised over admissible designs (Rainforth et al., 2023). In the setting of a PDE-governed inverse problem, each MCMC step necessitates at least one computation of the underlying PDE solution, meaning that the process quickly becomes computationally demanding (Alexande­rian, 2021). Traditional surrogate model approaches (Gramacy, 2020) are not readily applicable if the parameters are functional, or if the forward map exhibits non-Gaussian noise. In the latter case, the likelihood becomes intractable due to the introduction of auxiliary variables, and one must re­sort to expensive pseudo-marginal MCMC methods (Andrieu & Roberts, 2009), simulation based inference approaches (Glaser et al., 2022) or synthetic likelihoods (Price et al., 2018).

Our work offers a computationally efficient alternative to these methods by learning a generative model for the joint posterior distribution over $(a, \mathcal{G}(a))$, i.e., a joint distribution over the parameters of interest and the value of the (possibly stochastic) solution map. This extends the scope of classical surrogate methods, since no deterministic relationship between the parameters and solutions of the model is assumed. Crucially for BED, the proposed method models the association between the

parameter $a$ and the functional-form of the solution $\mathcal{G}(a)$, and thus is not dependent on a fixed set of evaluation points $\underline{\mathbf{x}}$ or a particular discretisation of the domain. To achieve this, we embed the functions in a finite-dimensions using implicit neural representations (Sitzmann et al., 2020) and use an energy-based model (Lecun et al., 2006) to model the distribution of latent representations.

## 2 ENERGY-BASED COUPLING ON FUNCTION SPACES

We propose to approximate the solution operator $\mathcal{G}$ using a probabilistic model $p_\theta$ over the joint distribution of $(a, \mathcal{G}(a))$ such that high likelihood regions of $p_\theta$ correspond to solutions $u = \mathcal{G}(a)$. To do so, we proceed in two steps: Using training data consisting of pairs $\{(a_i(\mathbf{x}_i^j), u_i(\mathbf{x}_i^j))_{j=1}^{N_i}\}_{i=1}^M$ with $u_i(\mathbf{x}_i^j) = \mathcal{G}(a_i)(\mathbf{x}_i^j)$, $\mathbf{x}_i^j \in \Omega$, we encode the functions $a$ and $u$ into finite-dimensional latent codes $\mathbf{z}_a, \mathbf{z}_u$. On the finite-dimensional representation space, we learn the joint distribution of the latent codes using a so-called energy-based model, where $p_\theta(\mathbf{z}_a, \mathbf{z}_u) \propto \exp(-E_\theta(\mathbf{z}_a, \mathbf{z}_u))$. Our workflow is visualised in Figure 4. Through the learnt latent representations, the model is independent of any pre-specified evaluation grid. This not only allows inference of solutions from observations at arbitrary points, but it also enables Bayesian experimental design of sensor placement as a downstream task.

**Learning the Implicit Neural Representations (INR)**     Following COIN++ (Dupont et al., 2022), we compress the functional data points $(a_i)_{i=1}^M$ and $(u_i)_{i=1}^M$ into implicit neural representations defined by $a_i(\cdot) = g_\psi(\cdot, \mathbf{z}_{a_i})$ and $u_i(\cdot) = f_\phi(\cdot, \mathbf{z}_{u_i})$. We generate training data by sampling parameters $a \in \mathcal{A}$ from a pre-specified prior distribution and simulating the forward dynamics $\mathcal{G}(a)$. We then train the neural representations by minimising the mean square error between the function value and the neural network prediction at random evaluation points. Each layer of $g_\psi$ and $f_\phi$ takes the form of a SIREN layer $\sin(\omega_0(W\mathbf{h} + \mathbf{b} + \boldsymbol{\beta}))$ (Sitzmann et al., 2020). While the weights and biases $W, \mathbf{b}$ of each layer are shared among all data points, the shifts $\boldsymbol{\beta}$ depend on the data point specific latent code $\mathbf{z}$ and are trained individually for each functional data point. This produces highly compressed latent representations $(\mathbf{z}_{a_i}, \mathbf{z}_{u_i})_{i=1}^M$, which can be decoded efficiently by applying the maps $g_\psi$ and $f_\phi$, respectively.

**Energy-Based Neural Coupling**     Energy-based models (EBMs) (Lecun et al., 2006) are unnormalised statistical models of the form $\exp(-E_\theta))$, where the energy-function $E_\theta$ is typically modelled with a scalar-valued neural network. We assume lossless compression in the INR and learn the joint distribution of $a$ and $u = \mathcal{G}(a)$ as an energy-based model over tuples $(\mathbf{z}_{a_i}, \mathbf{z}_{u_i})$. The architecture of the energy function first embeds the latent codes into an embedding space of shared dimension and processes the embedded vectors using MLPs. The suggested architecture is resilient to vanishing signals, ensuring that information flows in a stable manner (for details, see Appendix B.2). Since unnormalised models are not amenable to optimisation with maximum likelihood estimation, we explore training the model with contrastive divergence (Hinton et al., 2006) and energy discrepancy (Schröder et al., 2023), achieving the best results with the latter approach.

**Inference from sparse observations**     At inference time, we are given noisy observations of the system at finitely many evaluation points $\mathcal{D} = \{(\mathbf{x}_i, \mathbf{y}_i)\}_{i=1}^n$ with $\mathbf{y}_i = u(\mathbf{x}_i) + \eta_i$, where $\eta_i \sim \mathcal{N}(0, \sigma^2)$. The posterior distribution of the latent representations $(\mathbf{z}_a, \mathbf{z}_u)$ conditioned on the observed data is given by

$$p(\mathbf{z}_a, \mathbf{z}_u | \mathcal{D}) \propto p(\mathcal{D} | \mathbf{z}_a, \mathbf{z}_u) p_\theta(\mathbf{z}_a, \mathbf{z}_u) = \prod_{i=1}^n p(\mathbf{x}_i, y_i | \mathbf{z}_a, \mathbf{z}_u) p_\theta(\mathbf{z}_a, \mathbf{z}_u) \tag{1}$$

The noise assumption, together with the PDE model, results in $p(\mathbf{x}_i, \mathbf{y}_i | \mathbf{z}_a, \mathbf{z}_u) \approx \mathcal{N}(\mathbf{y}_i; f_\phi(\mathbf{x}_i, \mathbf{z}_u), \sigma^2)$. The desired parameter solution pair $(a, u = \mathcal{G}(a))$ can now be sampled from the posterior using stochastic gradient Langevin dynamics (Welling & Teh, 2011).

**Application to Bayesian experimental design**     One of the main motivations of our approach lies in Bayesian experimental design. Specifically, we seek to determine optimal sparse sensor placement positions $\underline{\mathbf{d}} = \{\mathbf{d}_1, \mathbf{d}_2, \ldots, \mathbf{d}_D\}$ for the inference of $(a, u = \mathcal{G}(a))$ based on $\mathbf{y} = u(\underline{\mathbf{d}}) + \boldsymbol{\eta}$. We measure the utility of a sensor placement position by calculating the expected information gain over the prior as measured by relative entropy, i.e. we use the utility function $U(\mathbf{d}) := \mathbb{E}_{p(\mathbf{y}|\mathbf{d})} D_{\text{KL}}(p(\mathbf{z}_a, \mathbf{z}_u | \mathbf{y}, \mathbf{d}) \| p(\mathbf{z}_a, \mathbf{z}_u))$. Using Bayes theorem and Monte Carlo estima-

tion we obtain the following biased utility estimator

$$\widetilde{U}(\mathbf{d}) = \frac{1}{K} \sum_{i=1}^{K} \log p(\mathbf{y}_i | \mathbf{z}_{a_i}, \mathbf{z}_{u_i}, \mathbf{d}) - \frac{1}{K} \sum_{i=1}^{K} \log \left( \frac{1}{M} \sum_{j=1}^{M} p(\mathbf{y}_i | \mathbf{z}_{a_j}, \mathbf{z}_{u_j}, \mathbf{d}) \right), \qquad (2)$$

where $(\mathbf{z}_{a_i}, \mathbf{z}_{u_i}), (\mathbf{z}_{a_j}, \mathbf{z}_{u_j}) \sim p(\mathbf{z}_a, \mathbf{z}_u)$ and $\mathbf{y}_i \sim \mathcal{N}(f_\phi(\mathbf{d}, \mathbf{z}_{u_i}), \sigma^2)$. The utility is sequentially optimised using Bayesian optimisation. For details, see Appendix C.

## 3 NUMERICAL EXPERIMENTS

Our training data consists of $M$ pairs of parameters and their corresponding solutions, $\{(a_i, u_i)\}_{i=1}^{M}$ for $a_i \in \mathcal{A}$ and $u_i \in \mathcal{U}$. We assume access to only $N_i$ point observations of them, where the set of $N_i$ locations varies across the $M$ function realizations and need not be the same for $a$ and $u$. While the method can handle functional parameters through the INR encoding, we assume for simplicity in the first presented example that $a$ is parameterised by a real-valued vector of finite dimension. We emphasise that PDE solutions are only required to train the INR and EBM models. Once trained, these models are reused for inference leading to high savings in terms of computational cost.

### 3.1 BOUNDARY VALUE PROBLEMS IN 1D

Consider the boundary value problem (BVP) on the interval $[-1, 1]$ given by the non-linear PDE

$$u''(x) - u^2(x)u'(x) = f(x), \quad u(-1) = X_a, \ u(1) = X_b,$$
$$f(x) = -\pi^2 \sin(\pi x) - \pi \cos(\pi x) \sin^2(\pi x),$$
$$X_a \sim \mathcal{N}(a, 0.3^2), \quad X_b \sim \text{Unif}(b - 0.3, b + 0.4), \quad a, b \sim \text{Unif}(-3, 3).$$

The training data consists of pairs $(a, b)$ and their corresponding solution for a realisation of $X_a$ and $X_b$. Figure 1 compares the numerical solutions of the BVP and samples from the learnt energy-based coupling. One can see that the sampled solutions resemble the numerical solutions and respect the boundary conditions. As a quantitative test we perform dimensionality reduction via t-SNE (van der Maaten & Hinton, 2008) which demonstrates that training data and generated data do not form clusters, i.e. the t-SNE test cannot distinguish between training and generated data.

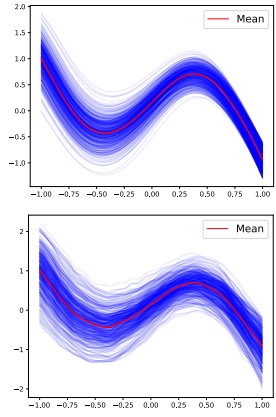

We perform inference on the parameters $a$ and $b$ when true values are set at random to $a = 2.9$ and $b = -0.11$, based on sparse observations of the PDE solution $\mathcal{D} = \{(x_i, y_i)\}_{i=1,\ldots,10}$, where $y_i = u(x_i) + \eta_i$, and $\eta_i$ are iid $\mathcal{N}(0, 0.1^2)$. Inference results for $10^3$ posterior samples from $p_\theta(\mathbf{z}_a, \mathbf{z}_u | \mathcal{D})$ are summarised in Figure 2. The left and right panels display histograms for sampled $a$ and $b$, respectively, while the central plot shows the generated solutions $f_\phi(\cdot, \mathbf{z}_u)$, together with the true solution and the observations $\mathcal{D}$. We observe a strong agreement between posterior samples and the ground truth. The relative $L^2$ error norm of the true solution and the posterior mean is $\|u - u_{\text{truth}}\|^2 / \|u_{\text{truth}}\|^2 = 0.065$ and the MSEs

Figure 1: Numerical simulations (top) and samples from the learnt coupling (bottom) conditional on parameters $(a, b) = (1, -1)$.

for $a$ and $b$ are 0.37 and 0.25, respectively. It is important to mention that we observe comparable values in all performance metrics regardless of the randomly chosen PDE parameters $(a, b)$ and the set of noisy observations $\mathcal{D}$.

### 3.2 STEADY-STATE DIFFUSION IN 2D

Next we consider learning the diffusion coefficient $\kappa$ of the 2D Darcy flow equation defined by the PDE $-\nabla \cdot \big(\kappa(\mathbf{x}) \nabla u(\mathbf{x})\big) = f(\mathbf{x})$ with domain $\mathbf{x} \in \Omega = [0, 1]^2$ and Dirichlet boundary conditions $u|_{\partial\Omega} = 0$. We assume that $\kappa$ and $f$ take the form

$$\kappa((x_1, x_2)) = \exp[(1.5 + a \cdot \cos(\pi x_1)(-1.5 + b \cdot \cos(\pi x_2)], \quad f = 0.5 + \varepsilon \ \text{ where } \ \varepsilon \sim \mathcal{N}(0, \sigma^2).$$

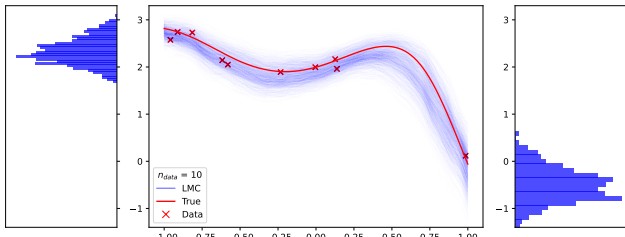

Figure 2: Inference results based on 10 noisy observations of a solution with $a = 2.9$ and $b = -0.11$. The central plot illustrates the solutions corresponding to $10^3$ samples from the posterior (blue), alongside the true solution (red). Left and right panels present histograms for $a$ and $b$, respectively.

Although the diffusion coefficient is characterised by just two parameters, we treat it as a function and learn an INR as outlined before. Using the learnt energy-based coupling, we perform an experimental design task in which, based on three initial observations (intentionally chosen in non informative places) of a solution with parameters $a = 1$ and $b = 1$, we maximise the estimated utility function to find optimal locations for ten additional measurement sites. To benchmark the efficacy of the new design locations, we generate the same number of points from a Sobol sequence (Sobol', 1967) within our domain and compare the posterior means along with the ground truth. We observe that results inferred from optimally chosen points closely match the ground truth unlike those inferred from Sobol points, see Figure 3. Corresponding plots for the diffusion coefficient can be found in the Appendix, Figure 6. In particular, the relative $L^2$ error norms between the ground truth and the posterior mean for the solution and the diffusion coefficient have lower values when performing inference based on optimal location measurements (see Table 1), with performance remaining constant over the entire domain of $a$ and $b$. It should be noted that the inference for Sobol points is also conducted using the learnt energy-based coupling.

| **Design points** | $\|\widehat{u} - u_{\mathrm{tr}}\|^2/\|u_{\mathrm{tr}}\|^2$ | $\|\log \widehat{\kappa} - \log \kappa_{\mathrm{tr}}\|^2/\|\log \kappa_{\mathrm{tr}}\|^2$ |
|---|---|---|
| BED with Energy-Based Coupling (Ours) | **0.021** | **0.013** |
| Sobol | 0.098 | 0.054 |

Table 1: Mean relative $L^2$ error norm for the solution and diffusion coefficient for BED experiment.



Figure 3: Left: Observed solution with initial observations (green crosses) and optimal design points (red crosses). Middle and right: posterior mean solutions based on optimal design points and Sobol points, respectively.

## 4 DISCUSSION AND RELATED WORK

The proposed approach is strongly related to the problem of *Neural Operator Learning*, a class of models that learn mappings between function spaces and solve partial differential equations. Examples include Fourier Neural Operators (Li et al., 2021) and Deep O-Nets (Lu et al., 2019). A common thread across these methods is their resolution invariance, i.e. their ability to generate predictions at any resolution. Existing work has largely focused on learning the deterministic relationship between the input and output, although recent work has sought to extend these approaches

to developing generative models on function space (Rahman et al., 2022; Lim et al., 2023). See also (Salvi et al., 2022). Our approach leverages similar resolution-invariant paradigms, integrating them into an EBM architecture to provide a generative associative map between function spaces. The results above already demonstrate the effectiveness of the proposed methodology, but a detailed exploration of the impact of different neural operator architectures is left for future work. These approaches holds promise for accelerating scientific discovery (Azizzadenesheli et al., 2024), such as enhancing the pathway to sustainable nuclear energy (Gopakumar et al., 2023; Kobayashi & Alam, 2024).

### ACKNOWLEDGMENTS

PCE is supported by EPSRC through the Modern Statistics and Statistical Machine Learning (StatML) CDT programme, grant no. EP/S023151/1. TS is supported by the EPSRC-DTP scholarship partially funded by the Department of Mathematics, Imperial College London. We thank the anonymous reviewer for their comments.

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

## A    DATASET DETAILS

**Boundary value problem**    The dataset consists of 20000 pairs of boundary conditions $(a, b)$ and associated solutions. Each solution is evaluated at $N_i$ points, where $N_i$ is a randomly chosen integer between 30 and 40.

**2D Darcy flow equation**    The dataset consists of 20000 pairs of diffusion coefficients and solutions of the PDE. Each solution and diffusion coefficient is evaluated at 784 points.

## B    IMPLEMENTATION DETAILS

We have implemented all experiments with PyTorch (Paszke et al., 2019).

### B.1    TRAINING

The training is done in two steps (as shown in Figure 4). First, we train the modulated INRs to represent the data. Once, the INRs have been fitted, we obtain the latent representations of the functions of interest. The latent modulations of the INRs are concatenated, in the cases where we seek for a prior over different functional parameters. Furthermore, as the data is encoded using only a few steps of gradient descent (for details, see Dupont et al. (2022)), the resulting standard deviation of the codes is very small, falling within the range of $[10^{-3}, 10^{-1}]$. Therefore, these raw latent representations are not appropriate for subsequent processing. To address this, we standardise the codes by subtracting the mean and dividing by the standard deviation. The normalised latent embeddings are then used to train the EBM model. The means and standard deviations are stored to unnormalised the samples generated by the EBM before using them as modulations for the INR to recover the functions in real space. It is important to remark that, unlike the INR representation, the use of the truncated coefficient expansion for a basis, such as the Fourier basis, to represent the function of interest does not scale well with dimension.

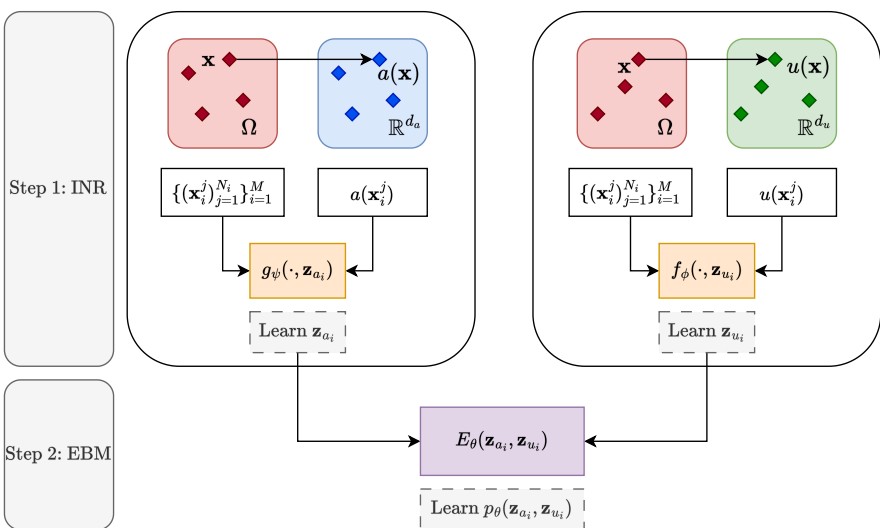

Figure 4: Workflow for the training of the joint INR and EBM model. Layout based on Serrano et al. (2023).

The final dimensions of the latent representations in the experiments outlined above are as follows

- Boundary value problem. Latent dimension for the solution function is 11.
- 2D Darcy flow equation. Latent dimensions for the solution function and diffusion coefficient are 16 and 2, respectively.

The criterion to determine the value of the latent dimension is that the mean relative $L^2$ error norm between the true function (either solution or parameter) and the one reconstructed from the latent embedding across all data points remains below a certain threshold. At the same time, we want the dimension to be low to ensure that the resulting distribution of the latent representations has a positive probability density that can be easily modelled by the EBM. For the diffusion coefficient, the relative $L^2$ error norm is of the order of $10^{-8}$ for a latent dimension of 2 and the error grows as the dimension increases or decreases. In particular, for a dimension greater than 2 the additional dimensions are redundant, with values repeating across dimensions. For the solutions the mean relative $L^2$ error norms for different latent dimensions computed on a validation set not seen during training for the BVP and the Darcy flow equation are shown in Tables 2 and 3, respectively.

Table 2: Mean relative $L^2$ error norm for the INR reconstructed solution of the BVP on a validation set.

| Dimension | Relative $L^2$ error norm |
|---|---|
| 15 | $6.1 \times 10^{-7}$ |
| 13 | $9.3 \times 10^{-7}$ |
| 11 | $1.1 \times 10^{-6}$ |
| 9 | $9.8 \times 10^{-6}$ |
| 7 | $8.7 \times 10^{-5}$ |

Table 3: Mean relative $L^2$ error norm for the INR reconstructed solution of the Darcy flow equation on a validation set.

| Dimension | Relative $L^2$ error norm |
|---|---|
| 64 | $7.7 \times 10^{-8}$ |
| 32 | $6.3 \times 10^{-5}$ |
| 16 | $5.5 \times 10^{-4}$ |
| 8 | $5.8 \times 10^{-3}$ |

## B.2 ARCHITECTURE DETAILS

### B.2.1 IMPLICIT NEURAL REPRESENTATION

We have only made minor changes to the SIREN architecture proposed by Sitzmann et al. (2020), so that it can take arbitrary point evaluations of the functions of interest and not just random points of a fixed grid. We have also followed their initialisation scheme.

### B.2.2 ENERGY-BASED MODEL

In all our experiments, each training point for the EBM consists of two parts, the PDE solution and a functional or vector-valued coefficient, and our goal is to understand the connection between the two by learning their joint probability density. To do this, we first uplift each part of the input vector into a latent space (using an encoder) so that they have the same dimension (equal to 128) and then propagate and merge them, with shared connections between the two branches. The architecture of the network is illustrated in Figure 5.

The specific structure of each element of the architecture is the following

- Encoder block

$$\text{Encoder}(\mathbf{x}) = \text{Linear}(\sigma \circ \text{Linear}(\boldsymbol{y}) + \boldsymbol{y}), \quad \boldsymbol{y} = \sigma \circ \text{Linear}(\mathbf{x}),$$

  where $\sigma$ is a GELU (Hendrycks & Gimpel, 2016) activation function.

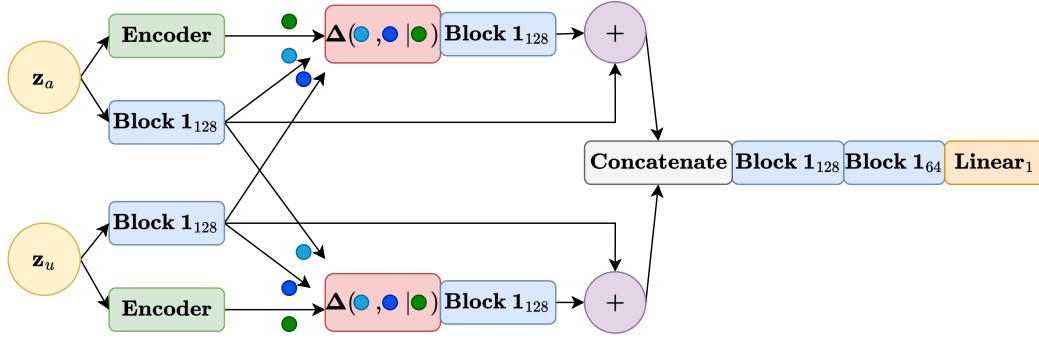

Figure 5: Energy-based model neural network architecture, where $\mathbf{z}_a$ and $\mathbf{z}_u$ represent the finite dimensional latent embeddings of two functions, such as, the solution and a coefficient of the PDE.

- Block 1
$$\text{Block } 1_k(\mathbf{x}) = \sigma \circ \text{Linear}(\mathbf{x}),$$
  where $\sigma$ is a RELU activation and $k$ denotes the output dimension.

- $\Delta$ operation
$$\Delta(\mathbf{x}, \mathbf{y}|\mathbf{z}) = (1 - \mathbf{z}) \cdot \mathbf{x} + \mathbf{z} \cdot \mathbf{y},$$
  where $\cdot$ denotes point-wise multiplication. Note that the vectors $\mathbf{x}, \mathbf{y}$ and $\mathbf{z}$ need to have the same dimension and the operation is just an interpolation.

- $+$ operation is a point-wise addition.

The dimension remains constant at 128 in the 2 branches of the architecture. Therefore, when both vectors are concatenated, it becomes 256. The last two Block 1's reduce the dimension from 256 to 128 and from 128 to 64, respectively.

## C  BED EXPERIMENT

As presented above, the experimental design task consists in optimally selecting $\mathbf{d}$ to maximise the information gain about the solution. In our proposed framework, this PDE solution is approximated by $f_\phi(\cdot, \mathbf{z}_u)$, where $\mathbf{z}_u$ is the associated latent embedding. Mathematically, the utility function for $\mathbf{d}$ needs to maximise the expected information gain over the prior $p(\mathbf{z}_a, \mathbf{z}_u)$, as measured by relative entropy. This is equivalent to maximising the expected KL-divergence

$$U(\mathbf{d}) = \mathbb{E}_{p(\mathbf{y}|\mathbf{d})}\left[D_{\text{KL}}\big(p(\mathbf{z}|\mathbf{d}, \mathbf{y}) \,\|\, p(\mathbf{z})\big)\right] = \int d\mathbf{z} \int d\mathbf{y}\big(\log p(\mathbf{z}|\mathbf{d}, \mathbf{y}) - \log p(\mathbf{z})\big) p(\mathbf{z}, \mathbf{y}|\mathbf{d}),$$

where the expectation is computed over the predictive distribution of the new (yet unobserved) data $p(\mathbf{y}|\mathbf{d})$ and for the sake of notation simplicity, we denote $\mathbf{z} = (\mathbf{z}_a, \mathbf{z}_u)$. Applying Bayes theorem, we rewrite the above expression in a form amenable to estimation

$$
\begin{aligned}
U(\mathbf{d}) &= \int d\mathbf{z} \int d\mathbf{y} \left( \log \frac{p(\mathbf{y}|\mathbf{z}, \mathbf{d})p(\mathbf{z})}{p(\mathbf{y}|\mathbf{d})} - \log p(\mathbf{z}) \right) p(\mathbf{z}, \boldsymbol{y}|\mathbf{d}) \\
&= \int d\mathbf{z} \int d\mathbf{y}\big(\log p(\mathbf{y}|\mathbf{z}, \mathbf{d}) - \log p(\mathbf{y}|\mathbf{d})\big) p(\mathbf{z}, \mathbf{y}|\mathbf{d}) \\
&= \mathbb{E}_{p(\mathbf{z}, \mathbf{y}|\mathbf{d})} \log p(\mathbf{y}|\mathbf{z}, \mathbf{d}) - \mathbb{E}_{p(\mathbf{y}|\mathbf{d})} \log p(\mathbf{y}|\mathbf{d}).
\end{aligned}
\tag{3}
$$

Notice that gradient-based optimisation of Eq. (3) with respect to $\mathbf{d}$ may suffer from high variance of the gradients arising from the need to differentiate through the probability density $p(\mathbf{z}, \mathbf{y}|\mathbf{d})$. This issue can be avoided by using Bayesian optimisation, wherein a differentiable surrogate function, such as a Gaussian process, is fitted to $U(\mathbf{d})$. Employing Monte Carlo estimation to compute the predictive distribution $p(\mathbf{y}|\mathbf{d})$, we derive the following estimator for the utility function

$$\hat{U}(\mathbf{d}) = \frac{1}{K}\sum_{i=1}^{K} \log p(\mathbf{y}_i|\mathbf{z}_i, \mathbf{d}) - \frac{1}{K}\sum_{i=1}^{K} \log \left( \frac{1}{M}\sum_{j=1}^{M} p(\mathbf{y}_i|\mathbf{z}_{i,j}, \mathbf{d}) \right),$$

where $\mathbf{z}_i, \mathbf{z}_{i,j} \sim p(\mathbf{z})$, $\mathbf{z}_i = (\mathbf{z}_{a_i}, \mathbf{z}_{u_i})$ and $\mathbf{y}_i \sim \mathcal{N}(f_\phi(\mathbf{d}, \mathbf{z}_{u_i}), \sigma^2)$. Assuming that the MCMC chains used for sampling from the posterior are ergodic, we can use the following estimator to reduce the computational cost

$$\tilde{U}(\mathbf{d}) = \frac{1}{K} \sum_{i=1}^{K} \log p(\mathbf{y}_i | \mathbf{z}_i, \mathbf{d}) - \frac{1}{K} \sum_{i=1}^{K} \log \left( \frac{1}{M} \sum_{j=1}^{M} p(\mathbf{y}_i | \mathbf{z}_j, \mathbf{d}) \right),$$

where the second term is evaluated using the numerically stabilised logsumexp function.

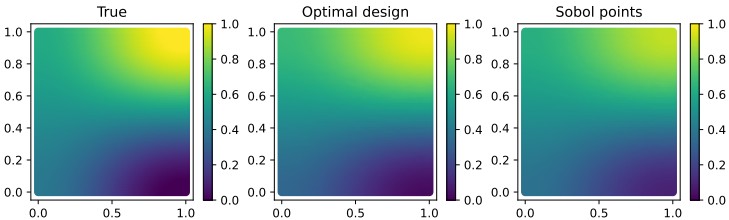

Figure 6: Left: normalised true diffusion coefficient, $\log \kappa$. Middle and right: posterior mean normalised diffusion coefficients, $\log \kappa$, based on optimal design points and Sobol points, respectively.

In the BED task, presented in Section 3.2, we have selected 10 optimal design points through Bayesian optimisation (Nogueira, 2014). This selection process was conducted sequentially, in the sense, that we choose new optimal measurement points one by one. This means that the prior of the next iteration is the posterior of the current iteration updated with the new observation of the function at the selected optimal point. Thus, the utility function we are trying to optimise is also updated at each iteration. We run the Bayesian optimisation algorithm for 50 steps in each iteration. At each step a Gaussian process is fitted to the known samples (points of the utility function previously explored), and the Gaussian process posterior, combined with a exploration strategy is used to determine the next point that should be explored.

The inference results of the experiment for the diffusion coefficient are shown in Figure 6. Similar to the results for the PDE solution, we observe that the posterior mean based on optimal design points is closer to the ground truth compared to that inferred from Sobol points.

