# OpenReview forum: "Optimal Experimental Design for Bayesian Inverse Problems using Energy-Based Couplings"
_ICLR.cc/2024/Workshop/AI4DiffEqtnsInSci — AI4DiffEqtnsInSci @ ICLR 2024 Poster_

### Official Review · Reviewer_U7L4 · 2024-02-23
**Bayesian experimental design using energy-based coupling and neural operators**

**Rating:** 5
**Confidence:** 3

**Review:**

Summary: The authors propose using implicit neural representations with energy-based coupling to provide an efficient surrogate for Bayesian experimental design.

Strengths and Weaknesses:

* It was slightly confusing to have Figure 2 come before Figure 1

[+] The paper is well-motivated and well-written. The application of neural operators to Bayesian experimental design (BED) is novel, and the authors make it clear why existing approaches would be inefficient compared to the proposed approach.

[-] Given the author's claim to prior inefficiencies for BED, i.e., having to solve the PDE for each MCMC step, it would be nice to see some baseline comparison highlighting how the author's approach is much faster and more efficient than prior approaches. I see no reporting of training or inference times.

[-] Does the author's INR model differ at all from CORAL (Serrano et. al., 2023)? It's stated the workflow is based on it, so I think the authors should state it by name, i.e., CORAL, like they do FNOs and DeepONets. This clarifies that the contribution is not CORAL itself, which has already been published. In the author's INR section, this is not referred to at all, while it is mentioned in the appendix, so it's a bit confusing what the authors are claiming as their contribution. This could be a misunderstanding of the author's modification to CORAL if that is the case, but I still think it should be named and clarified in any case.

[-] I fail to see a strong contribution with the problem solved in section 3.1. It appears the authors are just using the CORAL or, alternatively, FNO/DeepONet architectures, which they propose as future work, to perform inference. It seems to me the main contribution of the paper is the application of these function to function map neural architectures to BED. In the case of the problem in 3.2, this contribution is clearer, and the application to BED is shown to be better than Sobol points. The abstract is about this application to BED, so I fail to see how section 3.1 adds to that beyond standard operator learning results, albeit with energy-based coupling. Even in that case, the results of the learned coupling (using EBMs) have more spread and include high-frequency oscillations compared to the numerical simulations.

Conclusion: While the application of neural operators to BED is novel, the exact contributions and benefits of the author's approach are not clear and concise.

---

### Official Review · Reviewer_mKL1 · 2024-02-25
**The authors present and showcase an interesting idea for Bayesian Experimental Design, combining implicit neural representations and energy-based models with promising results.**

**Rating:** 9
**Confidence:** 4

**Review:**

Summary:
The work showcases a novel and intriguing approach to Bayesian Experimental Design (BED) aimed at overcoming computational barriers in inverse problems for partial differential equations (PDEs). The key idea is to learn implicit neural representations of the parameters $a$ and the respective solutions $u$ and to use an energy-based model to learn a joint distribution over these latent encodings. This yields an efficient and resolution-independent surrogate for the joint posterior distribution, enabling more effective optimization of experiments within the BED framework. It is a highly relevant problem with many applications in complex systems.

Pros:
- The manuscript is well-written, and the authors successfully provide condensed information on the broad range of concepts they bring together in their approach without compromising completeness.
- Relevant related work is discussed sufficiently.
- They neatly showcase the capabilities of their approach in two examples.

Cons:
- A comparison to some other more recent method would be desirable.

Further comments:
- It would help the reader if in the supplement B equation (3), the reformulation were spelled out.
- It would be helpful to explain briefly how the compressed latent representations are precisely retrieved from the COIN++ framework.
- As they are discussed in the text, the t-SNE plots should be added to the appendix.

---

### Official Review · Reviewer_YVdV · 2024-02-25
**Good approach to Bayesian experimental design coupling representations and energy-based model**

**Rating:** 8
**Confidence:** 3

**Review:**

Evaluation
-------------------------
**Quality:** The paper introduces a novel approach, Bayesian Experimental Design (BED), for optimizing experiments in the setting of inverse problems for partial differential equations (PDEs). It presents a thorough explanation of the proposed method, including mathematical formulations and implementation details. The approach is evaluated through numerical experiments, demonstrating its effectiveness in various scenarios. Additionally, the paper discusses the implications of the results and compares them with existing methods, providing insights into the strengths and limitations of the proposed approach. Overall, it is a high-quality work.

**Clarity:** The paper is generally well-written and organized, with clear sections outlining the problem statement, methodology, experiments, and discussion. Technical terms and concepts are defined and explained adequately, enhancing understanding for readers with varying levels of expertise. Figures and tables illustrate key points and present experimental results. However, some sections could benefit from additional clarification, particularly regarding the training process of implicit neural representations and energy-based models. This is not a problem for this short piece, especially because a very good supplementary material is provided for further clarification & details.

**Originality:** The paper introduces a novel approach to Bayesian Experimental Design for optimizing experiments in inverse problems for PDEs. It leverages implicit neural representations and energy-based models to model the joint posterior distribution efficiently, offering a unique perspective compared to existing simulation-based inference approaches. Integrating these techniques for experimental design in the context of PDEs is innovative, in my opinion, and extends the current SOTA in the field.

**Significance:** The proposed approach addresses significant computational barriers in optimizing experiments for inverse problems in PDEs, offering a computationally efficient alternative to existing methods. The potential applications of the proposed methodology are broad, spanning various domains where PDEs are used for modeling complex systems. Besides, the paper opens avenues for future research in neural operator learning and Bayesian experimental design, with implications for accelerating scientific discovery and solving real-world problems.

Pros
-------------------------
1. Introduces a novel approach to BED for optimizing experiments in the setting of inverse problems for PDEs.
2. Leverages implicit neural representations and energy-based models to model the joint posterior distribution efficiently.
3. Demonstrates effectiveness through numerical experiments on boundary value problems and diffusion equations in 1D and 2D.
4. Offers a computationally efficient alternative to existing simulation-based inference approaches.
5. Provides insights into the potential applications and future research directions in neural operator learning and BED

Cons
-------------------------
1. Some sections could benefit from additional clarification, particularly regarding the training process of implicit neural representations and energy-based models.
2. While the numerical experiments demonstrate effectiveness, further validation on more diverse datasets and real-world applications may be needed.
4. The computational cost of training implicit neural representations and energy-based models could be high, especially for larger-scale problems.

Typos and grammar issues
-------------------------
- "coupled random variables" - consider adding a hyphen for clarity ("coupled-random variables").
- "Equipping a with a prior distribution" ---> "Equipping 'a' with a prior distribution"
- "Each MCMC step necessitates at least one solve of the underlying PDE" - "solve" should be "solving"?
- "Design points kb" - unclear heading, needs clarification or correction.
- "experimental design task in which, based on three initial observations" --> "experimental design task involves using three initial observations"
- "results inferred from optimally chosen points are visually more similar to the ground truth" --> "results inferred from optimally chosen points closely match the ground truth"
- clean the bib; missing entries, capital letters. This is a nice tool btw: https://flamingtempura.github.io/bibtex-tidy/.

---

### Meta-Review · Area_Chair_gSui · 2024-03-01

**Recommendation:** Accept (Poster)

**Metareview:**

Authors propose using implicit neural representations coupled with energy-based models to provide an efficient surrogate for Bayesian experimental design. The high-level idea is interesting; I suggest authors make the following revisions to the paper for the camera-ready version. 1. clearly explain the contributions of the paper 2. better explain how the proposed method is different from prior work 3. provide more implementation details 4. quantify efficiency gains over baselines, and 5. demonstrating benefits on real-world problems for the camera-ready version.

---

### Decision · Program_Chairs · 2024-03-02

Accept (Poster)